# SampleRNN: An Unconditional End-to-End Neural Audio Generation Model

**Soroush Mehri**
University of Montreal

**Kundan Kumar**
IIT Kanpur

**Ishaan Gulrajani**
University of Montreal

**Rithesh Kumar**
SSNCE

**Shubham Jain**
IIT Kanpur

**Jose Sotelo**
University of Montreal

**Aaron Courville**
University of Montreal
CIFAR Fellow

**Yoshua Bengio**
University of Montreal
CIFAR Senior Fellow

## Abstract

In this paper we propose a novel model for unconditional audio generation based on generating one audio sample at a time. We show that our model, which profits from combining memory-less modules, namely autoregressive multilayer perceptrons, and stateful recurrent neural networks in a hierarchical structure is able to capture underlying sources of variations in the temporal sequences over very long time spans, on three datasets of different nature. Human evaluation on the generated samples indicate that our model is preferred over competing models. We also show how each component of the model contributes to the exhibited performance.

## 1 Introduction

Audio generation is a challenging task at the core of many problems of interest, such as text-to-speech synthesis, music synthesis and voice conversion. The particular difficulty of audio generation is that there is often a very large discrepancy between the dimensionality of the the raw audio signal and that of the effective semantic-level signal. Consider the task of speech synthesis, where we are typically interested in generating utterances corresponding to full sentences. Even at a relatively low sample rate of 16kHz, on average we will have 6,000 samples per word generated. [1]

Traditionally, the high-dimensionality of raw audio signal is dealt with by first compressing it into spectral or hand-engineered features and defining the generative model over these features. However, when the generated signal is eventually decompressed into audio waveforms, the sample quality is often degraded and requires extensive domain-expert corrective measures. This results in complicated signal processing pipelines that are to adapt to new tasks or domains. Here we propose a step in the direction of replacing these handcrafted systems.

In this work, we investigate the use of recurrent neural networks (RNNs) to model the dependencies in audio data. We believe RNNs are well suited as they have been designed and are suited solutions for these tasks (see Graves (2013), Karpathy (2015), and Siegelmann (1999)). However, in practice it is a known problem of these models to not scale well at such a high temporal resolution as is found when generating acoustic signals one sample at a time, e.g., 16000 times per second. This is one of the reasons that Oord et al. (2016) profits from other neural modules such as one presented by Yu & Koltun (2015) to show extremely good performance.

In this paper, an end-to-end unconditional audio synthesis model for raw waveforms is presented while keeping all the computations tractable.[2] Since our model has different modules operating at different clock-rates (which is in contrast to WaveNet), we have the flexibility in allocating the amount of computational resources in modeling different levels of abstraction. In particular, we can potentially allocate very limited resource to the module responsible for sample level alignments

---

[1] Statistics based on the average speaking rate of a set of TED talk speakers `http://sixminutes.dlugan.com/speaking-rate/`

[2] Code `https://github.com/soroushmehr/sampleRNN_ICLR2017` and samples `https://soundcloud.com/samplernn/sets`

operating at the clock-rate equivalent to sample-rate of the audio, while allocating more resources in modeling dependencies which vary very slowly in audio, for example identity of phoneme being spoken. This advantage makes our model arbitrarily flexible in handling sequential dependencies at multiple levels of abstraction.

Hence, our contribution is threefold:

1. We present a novel method that utilizes RNNs at different scales to model longer term dependencies in audio waveforms while training on short sequences which results in memory efficiency during training.

2. We extensively explore and compare variants of models achieving the above effect.

3. We study and empirically evaluate the impact of different components of our model on three audio datasets. Human evaluation also has been conducted to test these generative models.

## 2 SAMPLERNN MODEL

In this paper we propose SampleRNN (shown in Fig. 1), a density model for audio waveforms. SampleRNN models the probability of a sequence of waveform samples $X = \{x_1, x_2, \ldots, x_T\}$ (a random variable over input data sequences) as the product of the probabilities of each sample conditioned on all previous samples:

$$p(X) = \prod_{i=0}^{T-1} p(x_{i+1}|x_1, \ldots, x_i) \tag{1}$$

RNNs are commonly used to model sequential data which can be formulated as:

$$h_t = \mathcal{H}(h_{t-1}, x_{i=t}) \tag{2}$$
$$p(x_{i+1}|x_1, \ldots, x_i) = Softmax(MLP(h_t)) \tag{3}$$

with $\mathcal{H}$ being one of the known memory cells, Gated Recurrent Units (GRUs) (Chung et al., 2014), Long Short Term Memory Units (LSTMs) (Hochreiter & Schmidhuber, 1997), or their deep variations (Section 3). However, raw audio signals are challenging to model because they contain structure at very different scales: correlations exist between neighboring samples as well as between ones thousands of samples apart.

SampleRNN helps to address this challenge by using a hierarchy of modules, each operating at a different temporal resolution. The lowest module processes individual samples, and each higher module operates on an increasingly longer timescale and a lower temporal resolution. Each module conditions the module below it, with the lowest module outputting sample-level predictions. The entire hierarchy is trained jointly end-to-end by backpropagation.

### 2.1 FRAME-LEVEL MODULES

Rather than operating on individual samples, the higher-level modules in SampleRNN operate on *non-overlapping frames* of $FS^{(k)}$ ("Frame Size") samples at the $k^{\text{th}}$ level up in the hierarchy at a time (frames denoted by $f^{(k)}$). Each frame-level module is a deep RNN which summarizes the history of its inputs into a conditioning vector for the next module downward.

The variable number of frames we condition upon up to timestep $t-1$ is expressed by a fixed length hidden state or memory $h_t^{(k)}$ where $t$ is related to clock rate at that tier. The RNN makes a memory update at timestep $t$ as a function of the previous memory $h_{t-1}^{(k)}$ and an input $inp_t^{(k)}$. This input for top tier $k = K$ is simply the input frame. For intermediate tiers ($1 < k < K$) this input is a linear combination of conditioning vector from higher tier and current input frame. See Eqs. 4–5.

Because different modules operate at different temporal resolutions, we need to upsample each vector $c$ at the output of a module into a series of $r^{(k)}$ vectors (where $r^{(k)}$ is the ratio between the temporal resolutions of the modules) before feeding it into the input of the next module downward (Eq. 6). We do this with a set of $r^{(k)}$ separate linear projections.

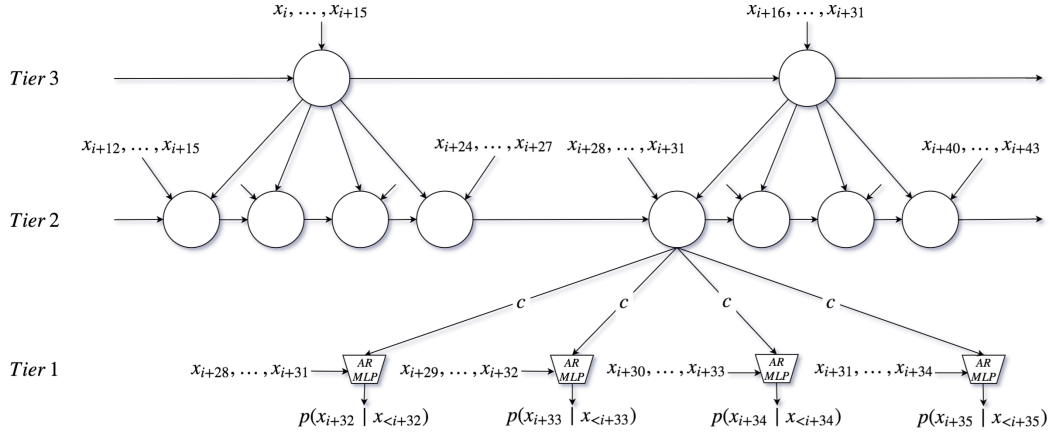

Figure 1: Snapshot of the unrolled model at timestep $i$ with $K = 3$ tiers. As a simplification only one RNN and up-sampling ratio $r = 4$ is used for all tiers.

Here we are formalizing the frame-level module in tier $k$. Note that following equations are exclusive to tier $k$ and timestep $t$ for that specific tier. To increase the readability, unless necessary superscript $(k)$ is not shown for $t$, $inp^{(k)}$, $W_x^{(k)}$, $h^{(k)}$, $\mathcal{H}^{(k)}$, $W_j^{(k)}$, and $r^{(k)}$.

$$inp_t = \begin{cases} W_x f_t^{(k)} + c_t^{(k+1)}; & 1 < k < K \\ f_t^{(k=K)}; & k = K \end{cases} \tag{4}$$

$$h_t = \mathcal{H}(h_{t-1}, inp_t) \tag{5}$$

$$c_{(t-1)*r+j}^{(k)} = W_j h_t; \qquad 1 \le j \le r \tag{6}$$

Our approach of upsampling with $r^{(k)}$ linear projections is exactly equivalent to upsampling by adding zeros and then applying a linear convolution. This is sometimes called "perforated" upsampling in the context of convolutional neural networks (CNNs). It was first demonstrated to work well in Dosovitskiy et al. (2016) and is a fairly common upsampling technique.

## 2.2 SAMPLE-LEVEL MODULE

The lowest module (tier $k = 1$; Eqs. 7–9) in the SampleRNN hierarchy outputs a distribution over a sample $x_{i+1}$, conditioned on the $FS^{(1)}$ *preceding samples* as well as a vector $c_i^{(k=2)}$ from the next higher module which encodes information about the sequence prior to that frame. As $FS^{(1)}$ is usually a small value and correlations in nearby samples are easy to model by a simple memoryless module, we implement it with a multilayer perceptron (MLP) rather than RNN which slightly speeds up the training. Assuming $e_i$ represents $x_i$ after passing through embedding layer (section 2.2.1), conditional distribution in Eq. 1 can be achieved by following and for further clarity two consecutive sample-level frames are shown. In addition, $W_x$ in Eq. 8 is simply used to linearly combine a frame and conditioning vector from above.

$$f_{i-1}^{(1)} = flatten([e_{i-FS^{(1)}}, \ldots, e_{i-1}]) \tag{7}$$

$$f_i^{(1)} = flatten([e_{i-FS^{(1)}+1}, \ldots, e_i])$$

$$inp_i^{(1)} = W_x^{(1)} f_i^{(1)} + c_i^{(2)} \tag{8}$$

$$p(x_{i+1}|x_1, \ldots, x_i) = Softmax(MLP(inp_i^{(1)})) \tag{9}$$

We use a Softmax because we found that better results were obtained by discretizing the audio signals (also see van den Oord et al. (2016)) and outputting a Multinoulli distribution rather than using a Gaussian or Gaussian mixture to represent the conditional density of the original real-valued signal. When processing an audio sequence, the MLP is convolved over the sequence, processing

each window of $FS^{(1)}$ samples and predicting the next sample. At generation time, the MLP is run repeatedly to generate one sample at a time. Table 1 shows a considerable gap between the baseline model RNN and this model, suggesting that the proposed hierarchically structured architecture of SampleRNN makes a big difference.

### 2.2.1 OUTPUT QUANTIZATION

The sample-level module models its output as a $q$-way discrete distribution over possible quantized values of $x_i$ (that is, the output layer of the MLP is a $q$-way Softmax).

To demonstrate the importance of a discrete output distribution, we apply the same architecture on real-valued data by replacing the $q$-way Softmax with a Gaussian Mixture Models (GMM) output distribution. Table 2 shows that our model outperforms an RNN baseline even when both models use real-valued outputs. However, samples from the real-valued model are almost indistinguishable from random noise.

In this work we use linear quantization with $q = 256$, corresponding to a per-sample bit depth of 8. Unintuitively, we realized that even linearly decreasing the bit depth (resolution of each audio sample) from 16 to 8 can ease the optimization procedure while generated samples still have reasonable quality and are artifact-free.

In addition, early on we noticed that the model can achieve better performance and generation quality when we *embed the quantized input values* before passing them through the sample-level MLP (see Table 4). The embedding steps maps each of the $q$ discrete values to a real-valued vector embedding. However, real-valued raw samples are still used as input to the higher modules.

### 2.2.2 CONDITIONALLY INDEPENDENT SAMPLE OUTPUTS

To demonstrate the importance of a sample-level autoregressive module, we try replacing it with "Multi-Softmax" (see Table 4), where the prediction of each sample $x_i$ depends only on the conditioning vector $c$ from Eq. 9. In this configuration, the model outputs an entire *frame* of $FS^{(1)}$ samples at a time, modeling all samples in a frame as conditionally independent of each other. We find that this Multi-Softmax model (which lacks a sample-level autoregressive module) scores significantly worse in terms of log-likelihood and fails to generate convincing samples. This suggests that modeling the joint distribution of the acoustic samples inside each frame is very important in order to obtain good acoustic generation. We found this to be true even when the frame size is reduced, with best results always with a frame size of 1, i.e., generating only one acoustic sample at a time.

### 2.3 TRUNCATED BPTT

Training recurrent neural networks on long sequences can be very computationally expensive. Oord et al. (2016) avoid this problem by using a stack of dilated convolutions instead of any recurrent connections. However, when they can be trained efficiently, recurrent networks have been shown to be very powerful and expressive sequence models. We enable efficient training of our recurrent model using *truncated backpropagation through time*, splitting each sequence into short subsequences and propagating gradients only to the beginning of each subsequence. We experiment with different subsequence lengths and demonstrate that we are able to train our networks, which model very long-term dependencies, despite backpropagating through relatively short subsequences.

Table 3 shows that by increasing the subsequence length, performance substantially increases alongside with train-time memory usage and convergence time. Yet it is noteworthy that our best models have been trained on subsequences of length 512, which corresponds to 32 milliseconds, a small fraction of the length of a single a phoneme of human speech while generated samples exhibit longer word-like structures.

Despite the aforementioned fact, this generative model can mimic the existing long-term structure of the data which results in more natural and coherent samples that is preferred by human listeners. (More on this in Sections 3.2–3.3.) This is due to the fast updates from TBPTT and specialized frame-level modules (Section 2.1) with top tiers designed to model a lower resolution of signal while leaving the process of filling the details to lower tiers.

## 3 EXPERIMENTS AND RESULTS

In this section we are introducing three datasets which have been chosen to evaluate the proposed architecture for modeling raw acoustic sequences. The description of each dataset and their preprocessing is as follows:

> **Blizzard** which is a dataset presented by Prahallad et al. (2013) for speech synthesis task, contains 315 hours of a single female voice actor in English; however, for our experiments we are using only 20.5 hours. The training/validation/test split is 86%-7%-7%.
>
> **Onomatopoeia**[3], a relatively small dataset with 6,738 sequences adding up to 3.5 hours, is human vocal sounds like grunting, screaming, panting, heavy breathing, and coughing. Diversity of sound type and the fact that these sounds were recorded from 51 actors and many categories makes it a challenging task. To add to that, this data is extremely unbalanced. The training/validation/test split is 92%-4%-4%.
>
> **Music** dataset is the collection of all 32 Beethoven's piano sonatas publicly available on `https://archive.org/` amounting to 10 hours of non-vocal audio. The training/validation/test split is 88%-6%-6%.

See Fig. 2 for a visual demonstration of examples from datasets and generated samples. For all the datasets we are using a 16 kHz sample rate and 16 bit depth. For the Blizzard and Music datasets, preprocessing simply amounts to chunking the long audio files into 8 seconds long sequences on which we will perform truncated backpropagation through time. Each sequence in the Onomatopoeia dataset is few seconds long, ranging from 1 to 11 seconds. To train the models on this dataset, zero-padding has been applied to make all the sequences in a mini-batch have the same length and corresponding cost values (for the predictions over the added 0s) would be ignored when computing the gradients.

We particularly explored two gated variants of RNNs—GRUs and LSTMs. For the case of LSTMs, the forget gate bias is initialized with a large positive value of 3, as recommended by Zaremba (2015) and Gers (2001), which has been shown to be beneficial for learning long-term dependencies.

As for models that take real-valued input, e.g. the RNN-GMM and SampleRNN-GMM (with 4 components), normalization is applied per audio sample with the global mean and standard deviation obtained from the train split. For most of our experiments where the model demands discrete input, binning was applied per audio sample.

All the models have been trained with teacher forcing and stochastic gradient decent (mini-batch size 128) to minimize the Negative Log-Likelihood (NLL) in bits per dimension (per audio sample). Gradients were hard-clipped to remain in [-1, 1] range. Update rules from the Adam optimizer (Kingma & Ba, 2014) ($\beta_1 = 0.9$, $\beta_2 = 0.999$, and $\epsilon = 1e{-}8$) with an initial learning rate of 0.001 was used to adjust the parameters. For training each model, random search over hyper-parameter values (Bergstra & Bengio, 2012) was conducted. The initial RNN state of all the RNN-based models was always learnable. Weight Normalization (Salimans & Kingma, 2016) has been used for all the linear layers in the model (except for the embedding layer) to accelerate the training procedure. Size of the embedding layer was 256 and initialized by standard normal distribution. Orthogonal weight matrices used for hidden-to-hidden connections and other weight matrices initialized similar to He et al. (2015). In final model, we found GRU to work best (slightly better than LSTM). 1024 was the the number of hidden units for all GRUs (1 layer per tier for 3-tier and 3 layer for 2-tier model) and MLPs (3 fully connected layers with ReLU activation with output dimension being 1024 for first two layers and 256 for the final layer before softmax). Also $FS^{(1)} = FS^{(2)} = 2$ and $FS^{(3)} = 8$ were found to result in lowest NLL.

### 3.1 WAVENET RE-IMPLEMENTATION

We implemented the WaveNet architecture as described in Oord et al. (2016). Ideally, we would have liked to replicate their model exactly but owing to missing details of architecture and hyper-parameters, as well as limited compute power at our disposal, we made our own design choices so that the model would fit on a single GPU while having a receptive field of around 250 milliseconds,

---

[3]Courtesy of Ubisoft

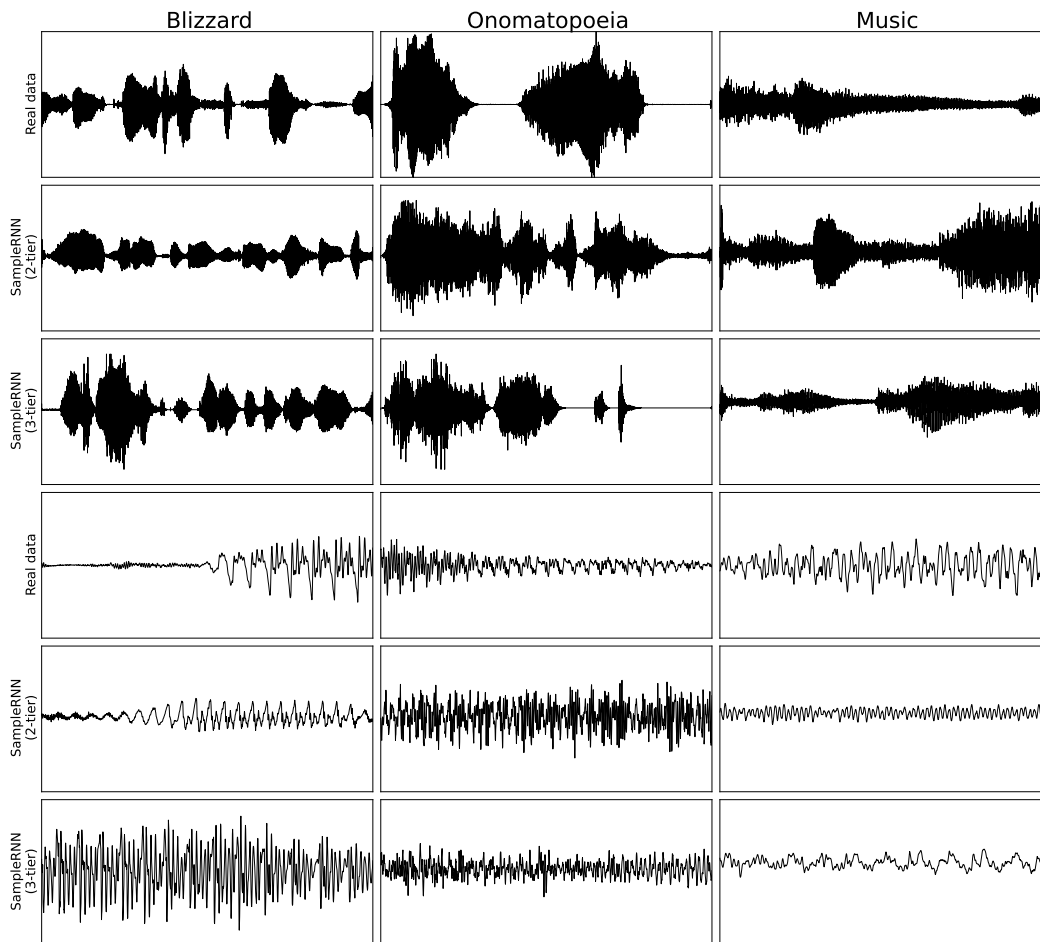

Figure 2: Examples from the datasets compared to samples from our models. In the first 3 rows, 2 seconds of audio are shown. In the bottom 3 rows, 100 milliseconds of audio are shown. Rows 1 and 4 are ground truth from which one can see how the datasets look different and have complex structure in low resolution which the frame-level component of the SampleRNN is designed to capture. Samples also to some extent mimic the same global structure. At the same time, zoomed-in samples of our model shows that it can perfectly resemble the high resolution structure present in the data as well.

Table 1: Test NLL in bits for three presented datasets.

| Model | Blizzard | Onomatopoeia | Music |
|---|---|---|---|
| RNN (Eq. 2) | 1.434 | 2.034 | 1.410 |
| WaveNet (re-impl.) | 1.480 | 2.285 | 1.464 |
| SampleRNN (2-tier) | 1.392 | 2.026 | **1.076** |
| SampleRNN (3-tier) | **1.387** | **1.990** | 1.159 |

Table 2: Average NLL on Blizzard test set for real-valued models.

| Model | Average Test NLL |
|---|---|
| RNN-GMM | -2.415 |
| SampleRNN-GMM (2-tier) | **-2.782** |

Table 3: Effect of subsequence length on NLL (bits per audio sample) computed on the Blizzard validation set.

| Subsequence Length | 32 | 64 | 128 | 256 | 512 |
|---|---|---|---|---|---|
| **NLL Validation** | 1.575 | 1.468 | 1.412 | 1.391 | 1.364 |

Table 4: Test (validation) set NLL (bits per audio sample) for Blizzard. Variants of SampleRNN are provided to compare the contribution of each component in performance.

| Model | NLL Test (Validation) |
|---|---|
| SampleRNN (2-tier) | 1.392 (1.369) |
| Without Embedding | 1.566 (1.539) |
| Multi-Softmax | 1.685 (1.656) |

while having a reasonable number of updates per unit time. Although our model is very similar to WaveNet, the design choices, e.g. number of convolution filters in each dilated convolution layer, length of target sequence to train on simultaneously (one can train with a single target with all samples in the receptive field as input or with target sequence length of size T with input of size receptive field + T - 1), batch-size, etc. might make our implementation different from what the authors have done in the original WaveNet model. Hence, we note here that although we did our best at exactly reproducing their results, there would very likely be different choice of hyper-parameters between our implementation and the one of the authors.

For our WaveNet implementation, we have used 4 dilated convolution blocks each having 10 dilated convolution layers with dilation 1, 2, 4, 8 up to 512. Hence, our network has a receptive field of 4092 acoustic samples i.e. the parameters of multinomial distribution of sample at time step t, $p(x_i) = f_\theta(x_{i-1}, x_{i-2}, \ldots x_{i-4092})$ where $\theta$ is model parameters. We train on target sequence length of 1600 and use batch size of 8. Each dilated convolution filter has size 2 and the number of output channels is 64 for each dilated convolutional layer (128 filters in total due to gated non-linearity). We trained this model using Adam optimizer with a fixed global learning rate of 0.001 for Blizzard dataset and 0.0001 for Onomatopoeia and Music datasets. We trained these models for about one week on a GeForce GTX TITAN X. We dropped the learning rate in the Blizzard experiment to 0.0001 after around 3 days of training.

## 3.2 HUMAN EVALUATION

Apart from reporting NLL, we conducted AB preference tests for random samples from four models trained on the Blizzard dataset. For unconditional generation of speech which at best sounds like mumbling, this type of test is the one which is more suited. Competing models were the RNN, SampleRNN (2-tier), SampleRNN (3-tier), and our implementation of WaveNet. The rest of the models were excluded as the quality of samples were definitely lower and also to keep the number of pair comparison tests manageable. We will release the samples that have been used in this test too.

All the samples were set to have the same volume. Every user is then shown a set of twenty pairs of samples with one random pair at a time. Each pair had samples from two different models. The human evaluator is asked to listen to the samples and had the option of choosing between the two model or choosing not to prefer any of them. Hence, we have a quantification of preference between every pair of models. We used the online tool made publicly available by Jillings et al. (2015).

Results in Fig. 3 clearly points out that SampleRNN (3-tier) is a winner by a huge margin in terms of preference by human raters, then SampleRNN (2-tier) and afterward two other models, which matches with the performance comparison in Table 1.

The same evaluation was conducted for Music dataset except for an additional filtering process of samples. Specific to only this dataset, we observed that a batch of generated samples from competing models (this time restricted to RNN, SampleRNN (2-tier), and SampleRNN (3-tier)) were either music-like or random noise. For all these models we only considered random samples that were not random noise. Fig. 4 is dedicated to result of human evaluation on Music dataset.

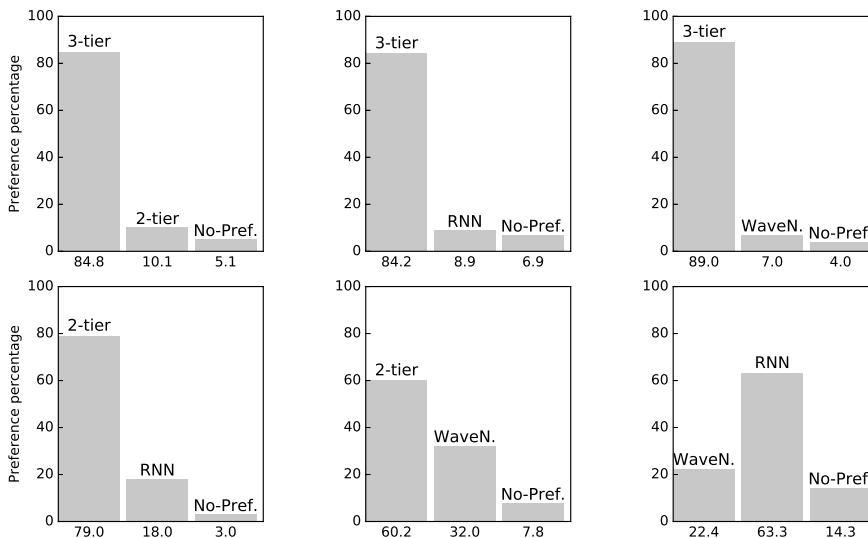

Figure 3: Pairwise comparison of 4 best models based on the votes from listeners conducted on samples generated from models trained on Blizzard dataset.

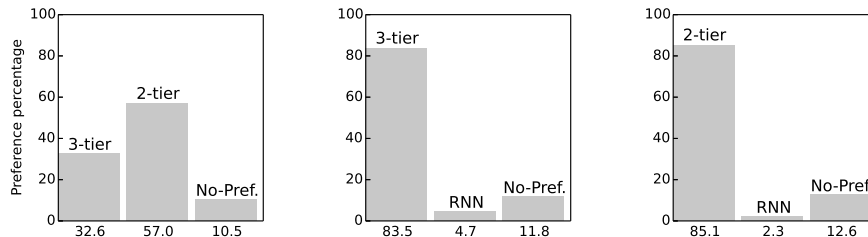

Figure 4: Pairwise comparison of 3 best models based on the votes from listeners conducted on samples generated from models trained on Music dataset.

## 3.3 QUANTIFYING INFORMATION RETENTION

For the last experiment we are interested in measuring the memory span of the model. We trained our model, SampleRNN (3-tier), with best hyper-parameters on a dataset of 2 speakers reading audio books, one male and one female, respectively, with mean fundamental frequency of 125.3 and 201.8Hz. Each speaker has roughly 10 hours of audio in the dataset that has been preprocessed similar to Blizzard. We observed that it learned to stay consistent generating samples from the same speaker without having any knowledge about the speaker ID or any other conditioning information. This effect is more apparent here in comparison to the unbalanced Onomatopoeia that sometimes mixes two different categories of sounds.

Another experiment was conducted to test the effect of memory and study the effective memory horizon. We inject 1 second of silence in the middle of sampling procedure in order to see if it will remember to generate from the same speaker or not. Initially when sampling we let the model generate 2 seconds of audio as it normally do. From 2 to 3 seconds instead of feeding back the generated sample at that timestep a silent token (zero amplitude) would be fed. From 3 to 5 seconds again we sample normally; feeding back the generated token.

We did classification based on mean fundamental frequency of speakers for the first and last 2 seconds. In 83% of samples SampleRNN generated from the same person in two separate segments.

This is in contrast to a model with fixed past window like WaveNet where injecting 16000 silent tokens (3.3 times the receptive field size) is equivalent to generating from scratch which has 50% chance (assuming each 2-second segment is coherent and not a mixed sound of two speakers).

## 4 RELATED WORK

Our work is related to earlier work on auto-regressive multi-layer neural networks, starting with Bengio & Bengio (1999), then NADE (Larochelle & Murray, 2011) and more recently PixelRNN (van den Oord et al., 2016). Similar to how they tractably model joint distribution over units of the data (e.g. words in sentences, pixels in images, etc.) through an auto-regressive decomposition, we transform the joint distribution of acoustic samples using Eq. 1.

The idea of having part of the model running at different clock rates is related to multi-scale RNNs (Schmidhuber, 1992; El Hihi & Bengio, 1995; Koutnik et al., 2014; Sordoni et al., 2015; Serban et al., 2016).

Chung et al. (2015) also attempt to model raw audio waveforms which is in contrast to traditional approaches which use spectral features as in Tokuda et al. (2013), Bertrand et al. (2008), and Lee et al. (2009).

Our work is closely related to WaveNet (Oord et al., 2016), which is why we have made the above comparisons, and makes it interesting to compare the effect of adding higher-level RNN stages working at a low resolution. Similar to this work, our models generate one acoustic sample at a time conditioned on all previously generated samples. We also share the preprocessing step of quantizing the acoustics into bins. Unlike this model, we have different modules in our models running at different clock-rates. In contrast to WaveNets, we mitigate the problem of long-term dependency with hierarchical structure and using stateful RNNs, i.e. we will always propagate hidden states to the next training sequence although the gradient of the loss will not take into account the samples in previous training sequence.

## 5 DISCUSSION AND CONCLUSION

We propose a novel model that can address unconditional audio generation in the raw acoustic domain, which typically has been done until recently with hand-crafted features. We are able to show that a hierarchy of time scales and frequent updates will help to overcome the problem of modeling extremely high-resolution temporal data. That allows us, for this particular application, to learn the data manifold directly from audio samples. We show that this model can generalize well and generate samples on three datasets that are different in nature. We also show that the samples generated by this model are preferred by human raters.

Success in this application, with a general-purpose solution as proposed here, opens up room for more improvement when specific domain knowledge is applied. This method, however, proposed with audio generation application in mind, can easily be adapted to other tasks that require learning the representation of sequential data with high temporal resolution and long-range complex structure.

ACKNOWLEDGMENTS

The authors would like to thank João Felipe Santos and Kyle Kastner for insightful comments and discussion. We would like to thank the Theano Development Team (2016)[4] and MILA staff. We acknowledge the support of the following agencies for research funding and computing support: NSERC, Calcul Québec, Compute Canada, the Canada Research Chairs and CIFAR. Jose Sotelo also thanks the Consejo Nacional de Ciencia y Tecnología (CONACyT) as well as the Secretaría de Educación Pública (SEP) for their support. This work was a collaboration with Ubisoft.

---

[4]http://deeplearning.net/software/theano/

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

## APPENDIX A

### A MODEL VARIANT: SAMPLERNN-WAVENET HYBRID

SampleRNN-WaveNet model has two modules operating at two different clock-rate. The slower clock-rate module (frame-level module) sees one frame (each of which has size *FS*) at a time while the faster clock-rate component(sample-level component) sees one acoustic sample at a time i.e. the ratio of clock-rates for these two modules would be the size of a single frame. Number of sequential steps for frame-level component would be *FS* times lower. We repeat the output of each step of frame-level component *FS* times so that number of time-steps for output of both the components match. The output of both these modules are concatenated for every time-step which is further operated by non-linearities for every time-step independently before generating the final output.

In our experiments, we kept size of a single frame (*FS*) to be 128. We tried two variants of this model: 1. fully convolutional WaveNet and 2. RNN-WaveNet. In fully convolutional WaveNet, both modules described above are implemented using dilated convolutions as described in original WaveNet model. In RNN-WaveNet, we use high capacity RNN in the frame-level module to model the dependency between frames. The sample-level WaveNet in RNN-WaveNet has receptive field of size 509 samples from the past.

Although these models are designed with the intention of combining the two models to harness their best features, preliminary experiments show that this variant is not meeting our expectations at the moment which directs us to a possible future work.

