# Peer review of "SampleRNN: An Unconditional End-to-End Neural Audio Generation Model"

_ICLR 2017 — accepted_

[Author Response · Soroush Mehri · 07 Nov 2016 (modified: 06 Dec 2016)]
**Code and generated samples**

Following is a link to the generated samples from our model and baseline models:

[Public Comment · Aleksandar Botev · 06 Dec 2016]
**Lack of mathematical formalism all over**

The paper looks very nice and the results are even better. However, I must clearly state that the way that you have defined your model lacks any mathematical formalism. The main definition in the paper is literally Figure 1, with literally 2 equations in the whole paper, where one of them is just defining an ordered cascade definition of a joint distribution (eq 1). It is stated that each of the modules operates on a different FS_k time window, but how are these combined with the previous h-vector and with the one coming from the upper modules is not stated and remains a mystery, unless I go and read your code. Also interestingly enough the lowest level module uses an MLP rather than some form of an RNN to combine the observation in its window and the conditioning vector c. I'm very confused on why is that the case, and did you just stack all of the FS_1 vectors and input them to the MLP? Did you try to have another RNN there rather than an MLP? Does this implies that at every new iteration t, we need to compute the MLP at every level a new? 

Also your answer to Soroush Mehri 3) - "You can find the code and best found hyper-parameters in the following Github repo" is really unsatisfactory. As a researcher I will need to spent several hours digging trough your code, written in Theano, which for that matter many people might have never used before, in order to just understand what exactly are you doing. On the other hand you can just explain in probably 3 equations the whole model and another 2 sentences about what were the number of hidden units etc... Also speaking about research, you should NOT just report the best chosen hyper parameters. The whole community will benefit a lot more if you list all of the things you tried and how well did they do (if needed add this in an Appendix). 

I really hope you guys take this constructively and write at least the model definition better.

[Official Review · AnonReviewer2 · rating 8 · confidence 3 · 12 Dec 2016]
**Promising work, paper lacking details**
soundness 3 · originality 2 · clarity 3 · impact 4

Pros:
The authors are presenting an RNN-based alternative to wavenet, for generating audio a sample at a time.
RNNs are a natural candidate for this task so this is an interesting alternative. Furthermore the authors claim to make significant improvement in the quality of the produces samples.
Another novelty here is that they use a quantitative likelihood-based measure to assess them model, in addition to the AB human comparisons used in the wavenet work.

Cons:
The paper is lacking equations that detail the model. This can be remedied in the camera-ready version.
The paper is lacking detailed explanations of the modeling choices:
- It's not clear why an MLP is used in the bottom layer instead of (another) RNN.
- It's not clear why r linear projections are used for up-sampling, instead of feeding the same state to all r samples, or use a more powerful type of transformation. 
As the authors admit, their wavenet implementation is probably not as good as the original one, which makes the comparisons questionable. 

Despite the cons and given that more modeling details are provided, I think this paper will be a valuable contribution.

[Official Review · AnonReviewer3 · rating 8 · confidence 4 · 16 Dec 2016]
**No Title**
soundness 5 · originality 2 · appropriateness 4 · recommendation (unofficial) 4

The paper introduces SampleRNN, a hierarchical recurrent neural network model of raw audio. The model is trained end-to-end and evaluated using log-likelihood and by human judgement of unconditional samples, on three different datasets covering speech and music. This evaluation shows the proposed model to compare favourably to the baselines.

It is shown that the subsequence length used for truncated BPTT affects performance significantly, but interestingly, a subsequence length of 512 samples (~32 ms) is sufficient to get good results, even though the features of the data that are modelled span much longer timescales. This is an interesting and somewhat unintuitive result that I think warrants a bit more discussion.

The authors have attempted to reimplement WaveNet, an alternative model of raw audio that is fully convolutional. They were unable to reproduce the exact model architecture from the original paper, but have attempted to build an instance of the model with a receptive field of about 250ms that could be trained in a reasonable time using their computational resources, which is commendable.

The architecture of the Wavenet model is described in detail, but it found it challenging to find the same details for the proposed SampleRNN architecture (e.g. which value of "r" is used for the different tiers, how many units per layer, ...). I think a comparison in terms of computational cost, training time and number of parameters would also be very informative.

Surprisingly, Table 1 shows a vanilla RNN (LSTM) substantially outperforming this model in terms of likelihood, which is quite suspicious as LSTMs tend to have effective receptive fields of a few hundred timesteps at best. One would expect the much larger receptive field of the Wavenet model to be reflected in the likelihood scores to some extent. Similarly, Figure 3 shows the vanilla RNN outperforming the Wavenet reimplementation in human evaluation on the Blizzard dataset. This raises questions about the implementation of the latter. Some discussion about this result and whether the authors expected it or not would be very welcome.

Table 1 and Figure 4 also show the 2-tier SampleRNN outperforming the 3-tier model in terms of likelihood and human rating respectively, which is very counterintuitive as one would expect longer-range temporal correlations to be even more relevant for music than for speech. This is not discussed at all, I think it would be useful to comment on why this could be happening.

Overall, this an interesting attempt to tackle modelling very long sequences with long-range temporal correlations and the results are quite convincing, even if the same can't always be said of the comparison with the baselines. It would be interesting to see how the model performs for conditional generation, seeing as it can be more easily be objectively compared to models like Wavenet in that domain.



Other remarks:

- upsampling the output of the models is done with r separate linear projections. This choice of upsampling method is not motivated. Why not just use linear interpolation or nearest neighbour upsampling? What is the advantage of learning this operation? Don't the r linear projections end up learning largely the same thing, give or take some noise?

- The third paragraph of Section 2.1.1 indicates that 8-bit linear PCM was used. This is in contrast to Wavenet, for which an 8-bit mu-law encoding was used, and this supposedly improves the audio fidelity of the samples. Did you try this as well?

- Section 2.1 mentions the discretisation of the input and the use of a softmax to model this discretised input, without any reference to prior work that made the same observation. A reference is given in 2.1.1, but it should probably be moved up a bit to avoid giving the impression that this is a novel observation.

[Official Review · AnonReviewer1 · rating 9 · confidence 4 · 17 Dec 2016]
**No Title**
soundness 4 · meaningful comparison 3

The paper proposed a novel SampleRNN to directly model waveform signals and achieved better performance both in terms of objective test NLL and subjective A/B tests. 

As mentioned in the discussions, the current status of the paper lack plenty of details in describing their model. Hopefully, this will be addressed in the final version.

The authors attempted to compare with wavenet model, but they didn't manage to get a model better than the baseline LSTM-RNN, which makes all the comparisons to wavenets less convincing. Hence, instead of wasting time and space comparing to wavenet, detailing the proposed model would be better.

[Author Response · Soroush Mehri · 19 Jan 2017]
**Changelog**

Modifications to the last revision:
- Section 2, SampleRNN model. Better model description, changed order of sub-sections, explained the upsampling method more clearly.
- Added a paragraph (before Section 3.1) detailing the training procedure and hyper-parameters.
- Added subsection 3.3 Quantifying information retention. (See authors' response to AnonReviewer3 on Dec. 2 titled "time horizon")
- Minor modifications (typo, citation, rephrasing, etc.)

[Final Decision · Program Chairs · 06 Feb 2017]
**ICLR committee final decision**

The reviewers were unanimous in their agreement about accepting this paper.
 Pros 
 - novel formulation that don't require sample by sample prediction
 - interesting results
 
 Cons
 - lack of details / explanation in the mathematical formulation / motivations for the model.